# Myocardial Work Does Not Have Additional Diagnostic Value in the Assessment of ATTR Cardiac Amyloidosis

**DOI:** 10.3390/jcm10194555

**Published:** 2021-09-30

**Authors:** Michael Y. Henein, Per Lindqvist

**Affiliations:** 1Heart Centre, Department of Cardiology, Umeå University, 90585 Umeå, Sweden; michael.henein@umu.se; 2Institute of Public Health and Clinical Medicine, Umeå University, 90585 Umeå, Sweden; 3Heart Centre, Department of Clinical Physiology, Umeå University, 90585 Umeå, Sweden; 4Institute of Surgical and Perioperative Sciences, Umeå University, 90585 Umeå, Sweden

**Keywords:** cardiac amyloidosis, echocardiography, DPD scintigraphy, prognosis

## Abstract

Background: Reduced LV longitudinal strain (GLS) and increased relative apical sparing (RELAPS) and increased wall thickness have been proposed as features for transthyretin cardiac amyloidosis (ATTR-CA). Myocardial work (MW) has recently been shown as useful afterload independent disease marker, hence we aimed to investigate its use in differentiating ATTR-CA from heart failure with increased septal thickness but no cardiac amyloidosis (CA) (HFnCA). Methods: This study included patients with HF and increased septal thickness ≥ 14 mm. We included 59 patients with hereditary (ATTRv) and 27 wild type transthyretin amyloidosis (ATTRwt) described as ATTR-CA based on DPD scintigraphy. We also enrolled 30 non-amyloidosis heart failure patients with negative DPD scintigraphy, as a control group. Myocardial work (MW) was used to assess the index (GWI), constructive (GCW) and wasted (GWW) work. Relative wall thickness (RWT) and relative apical sparing (RELAPS) were tested as conventional measures. Results: The RWT and RELAPS were higher in ATTR-CA (*p* < 0.001) and predicted ATTR-CA (RWT; AUC = 0.84, *p* < 0.001) and RELAPS (AUC = 0.81, *p* < 0.001). MW; GWI (*p* = 0.04), GCW (*p* = 0.03), GWW (*p* = 0.001) were all lower in ATTR-CA compared with HFnCA but only GWW predicted ATTR-CA, (AUC = 0.75, *p* < 0.001). Binary logistic univariate regression analysis showed RWT (*p* = 0.003, *β* = 16.2) and RELAPS (*p* = 0.003, *β* = 2.3) to be associated with ATTR-CA but not MW. GWI and GCW correlated with NT-proBNP (*p* < 0.05) and Troponin (*p* < 0.01), but not RWT or RELAPS. Conclusion: Myocardial work had lower accuracy, compared to RWT or RELAPS, in identifying ATTR-CA but was better related to biomarkers. Thus, MW assessment is unlikely to have additional value in improving the diagnosis of heart failure due to ATTR-CA.

## 1. Introduction

Transthyretin (TTR) amyloidosis (ATTR) is the most common type of systemic amyloidosis and is classified into a hereditary type (ATTRv), and a wild type transthyretin (ATTRwt) amyloidosis [1,2]. Typical cardiac features that raise the suspicion of ATTR-CA can be obtained from echocardiographic examination, although definitive diagnosis is always confirmed either by myocardial biopsy or non-invasively using nuclear bone tracer scintigraphy [3,4]. Typical echocardiographic findings suggestive of ATTR-CA include increased left ventricular (LV) wall thickness followed by development of restrictive ventricular filling pattern [5] due to loss of cavity compliance. Recently, myocardial deformation in the form of reduced global LV longitudinal strain (GLS) and relative apical sparing (RELAPS) have been proposed as additional echocardiographic features for ATTR-CA [6,7]. In addition, myocardial work (MW) has been proposed as additional marker predicting prognosis in cardiac amyloidosis [8]. However, no study has, so far, tested the use of MW in identifying ATTR-CA in patients with heart failure and increased wall thickness. 

This study has two clear objectives, firstly to investigate the use of MW in ATTR-CA and test its accuracy in differentiating patients with cardiac ATTR from those with heart failure and increased wall thickness but no cardiac amyloidosis (CA) (HFnCA) and Secondly, to test the accuracy of MW in predicting clinical prognosis in ATTR-CA.

## 2. Material and Methods

### 2.1. Patient Population

This is a retrospective study in patients with heart failure and increased LV septal thickness.

Inclusion criteria for further investigations were: (1) patients with IVSd ≥ 14 mm, (2) absence of types of CA other than ATTRwt and (3) confirmed final diagnosis of ATTR-CA using 99mTc-3,3-diphosphono-1,2-propanodicarboxylic acid (DPD) scintigraphy examination (grade 2-3). Based on these criteria, 27 patients were identified and included in this study. Part of this cohort findings was recently published [9]. In addition, 59 patients with ATTRv were included from our local database, studied between 2013–2020, all diagnosed with ATTR-CA based septal thickness ≥ 14mm and positive DPD scintigraphy (grade 2-3).

For comparison of disease progression and clinical prognosis, we also enrolled a control group of 30 non-amyloidosis heart failure patients (HFnCA), who all had septal thickness ≥ 14 mm but negative DPD scintigraphy examinations. Patients with Perugini score 1 were excluded from the study due to an uncertain diagnosis.

AL amyloidosis was excluded in all patients, based on blood and urine analysis of serum free light chain (FLC) abnormalities (Freelite, Binding Site reagent, reference range 0.27–1.64) and the presence of monoclonal bands. Patients with abnormalities in these analyses were carefully evaluated and clinical history and disease progression reviewed to assess the possibility of them having AL amyloidosis. Sequencing of the TTR-gene was also undertaken in all patients to diagnose ATTRv amyloidosis.

### 2.2. Echocardiographic Examination

Echocardiographic examination was performed using a Vivid E9 system (GE Medical Systems, Horten, Norway) equipped with an adult 1.5–4.3 MHz phased array transducer. All echocardiograms were performed by the same investigator (PL). Standard views from the parasternal long axis-, short axis- and apical long, four and two chamber views were used. Flow velocities were obtained using pulsed and continuous wave Doppler techniques as proposed by recent guidelines [10,11]. All acquisitions were made from the left lateral position.

Relative ventricular wall thickness (RWT) was calculated, according to the American society of echocardiography (ASE) recommendations, as posterior wall thickness [PWT]/Left ventricular (LV) diastolic diameter [LVDd]. LV mass was calculated using the Devereux formula [12]. Trans-mitral blood flow velocities were acquired with the sample volume placed at the tips of the mitral valve leaflets with optimal angulation to LV inflow. From trans-mitral flow velocities, we measured early (E) and atrial diastolic (A) flow velocities. Retrograde systolic trans-tricuspid flow was obtained from either parasternal right ventricular inflow or apical 4-chamber view, for measuring peak retrograde trans-tricuspid pressure drop using continuous wave Doppler, which reflects pulmonary artery systolic pressures. Left atrial volume was measured using the modified Simpson’s biplane calculation (4 and 2 chamber views).

Pulsed wave tissue Doppler analysis was also performed to assess mean LV lateral and septal myocardial early diastolic velocities (e’) at the mitral annular level and E/e’ was calculated [13]. All Doppler recordings were obtained at a sweep speed of 50–100 mm/s with a superimposed ECG (lead II). Off-line analyses were made using commercially available software (General Electric, EchoPac version BT 13, 113.0, Waukesha, WI, USA), and the means of three consecutive cardiac cycles were calculated.

### 2.3. Assessment of Myocardial Work

Anatomical landmarks were used, and care was taken for echocardiographic image acquisition to ensure adequate LV tracking, avoiding foreshortening of LV cavity when measuring global LV strain, all according to recent recommendations [14].

Longitudinal myocardial deformation was assessed from the 2-dimensional echocardiographic acquisitions using speckle tracking technology and was analyzed off-line. From the apical four chamber-, two chamber and apical parasternal long axis views the endocardial border detection of the septal, apical and lateral LV walls was made automatically and manually adjusted, in order to analyze global LV strain (GLS) measurements. GLS was measured at end-systole with the reference point set at the onset of two consecutive QRS-complexes of the superimposed ECG. Relative apical sparing (RELAPS) was also calculated as average apical strain/(average basal strain + average mid strain). To assess MW, systolic and diastolic blood pressure were inserted, and pressure-strain curve obtained in time according to aortic and mitral valve closure and opening time. Blood pressure was taken using conventional cuff mercury manometer after the echocardiographic examination and a mean of three consecutive measurements was taken. From pressure-strain curves, strain analyses were measured using a dedicated workstation (General Electric, EchoPac version BT 14, 114.0, Waukesha, WI, USA) and the following parameters calculated, Figure 1.
Global Work Index (GWI): total work within the area of the LV GLS = area within the pressure-strain curve (mmHg/%)Global Constructive Work (GCW): work performed by the LV contributing to cavity ejection. Constructive MW is defined as shortening of the myocytes during systole added to lengthening of the myocytes during isovolumic relaxation.Global Wasted Work (GWW): work performed by the LV that does not contribute to cavity ejection. Wasted MW is defined as lengthening of myocytes (rather than shortening) during systole added to shortening during the isovolumic relaxation phase.Global Work Efficiency (GWE): constructive MW/(constructive MW + wasted MW) (these values will not be affected by peak LV pressure).

### 2.4. 3, 3-diphosphono-1, 2-propanodicarboxylic Acid (DPD) Scintigraphy

All patients underwent DPD scintigraphy examination using an Infinia Hawkeye hybrid single-photon-emission computed-tomography gamma camera (SPECT-CT) (General Electric Medical Systems, Milwaukee, WI, USA) with a low-energy high-resolution collimator. An intravenous injection of ~750 MBq DPD was given 3 h prior to the acquisition of whole-body planar image, followed by a non-contrast, low dose CT scan and a SPECT acquisition, 60 projections and reconstructed to a 128 × 128 matrix (OSEM, 3 iterations, 10 subsets) with scatter and CT-based attenuation correction. Reconstruction of SPECT images was performed on the Xeleris workstation (GE Healthcare, Waukesha, WI, USA). DPD scores were reported by 2 experienced clinicians using the Perugini’s grading system [15] with grade 0 being negative and grades 1 to 3 increasingly positive.

### 2.5. Statistical Analysis

Statistical analysis was preformed using SPSS^®^, version 27 (IBM Corp., Armonk, NY, USA). Parametric data are presented as mean, standard deviation (SD), maximum and minimum values for continuous variables. Percentages were used to describe categorical variables. Categorical variables were compared using Chi-square and McNemar tests, and continuous variables were compared using the Student’s *t*-tests for normally distributed data or Mann-Whitney U tests for non-normally distributed data. Non-parametric data was presented as median and interquartile range 25–75. Normality was assessed by Shapiro-Wilk’s test. *p*-values were presented with a 0.05 level of statistical significance.

### 2.6. Study Ethics

This study complies with the declaration of Helsinki and the study protocol was approved by the Regional Ethics Committee of Umeå (reference numbers: 2016-435-31M, 2018-418-32M, 2018-137-32M). All subjects gave written informed consent to participate in the study.

## 3. Results

Patient characteristics are presented in Table 1. Eighty-six patients with ATTR-CA (mean age 77 ± 8 years, 46 females) and 30 control patients with heart failure (HFnCA) (mean age 77 ± 8 years, 11 females) were enrolled in the study. All patients and controls had septal thickness ≥ 14mm. The HFnCA patients had different etiologies for heart failure including coronary heart disease (56%), hypertensive heart disease (96%), aortic stenosis related heart failure (16%), and suspected hypertrophic cardiomyopathy (20%). Neither diagnosis of hypertrophic cardiomyopathy nor hypertrophic obstructive cardiomyopathy was confirmed by myocardial biopsy or MRI. At the time of DPD, ATTR-CA patient’s had higher NT-proBNP (*p* = 0.007), lower systolic and diastolic blood pressure (*p* = 0.02 and *p* = 0.004, respectively) and lower weight (*p* = 0.015) compared with the no ATTR-CA patients.

Echocardiographic data are shown in Table 2. In ATTR-CA heart rate was higher, septum and posterior walls and RWT thicker and LV diastolic dimension lower.

### 3.1. Echocardiographic Predictors of DPD Verified ATTR-CA

Relative wall thickness and RELAPS were higher in ATTR-CA (*p* < 0.001) and strongly predicted the diagnosis of ATTR-CA by DPD (RWT; AUC = 0.84, *p* < 0.001 and RELAPS; AUC = 0.81, *p* < 0.001, (Figure 2). In ATTR-CA, GWI (*p* = 0.04), GCW (*p* = 0.03), GWW (*p* = 0.001) were all lower than in HFnCA. The difference between groups disappeared after excluding patients with atrial fibrillation (Table 3).

On ROC analysis, GWW predicted ATTR-CA, AUC = 0.75, *p* < 0.001 but GCW did not (AUC = 0.62, *p* = 0.092) (Figure 2). The binary logistic univariate regression analysis showed RWT (*p* = 0.003, *β* = 16.2) and RELAPS (*p* = 0.003, *β* = 2.3) to be associated with ATTR-CA but not GCW or GWW (*p* > 0.05). MW quantification was feasible in 24 patients with HFnCA (86%) and in 53 of ATTR-CA patients (63%).

### 3.2. Myocardial Work and Biomarkers

In ATTR-CA patients, GLS GWI and GCW correlated negatively with NT-proBNP (*p* < 0.05), but only modestly. The same variables were more strongly correlated with Troponin (*p* < 0.01). RWT and RELAPS were not related to either BNP or Troponin, Table 4.

### 3.3. Myocardial Work and All Cause Mortality in ATTR-CA

In ATTR-CA, RWT, GLS, GWI, GCW and GWE were all lower in patients who died compared to survivors (Table 5). However, on univariate binary logistic regression, none of those variables was close to significance to enter in a model predicting all-cause mortality.

### 3.4. Variability Analysis

Inter-observer intraclass correlation (ICC) using tri-plane GLS was tested in 10 patients with ATTR-CA. The ICC was 0.96, *p* < 0.001, indicating excellent agreement.

## 4. Discussion

Findings: Our results show that MW was lower in ATTR-CA compared to HFnCA but was less accurate, compared with RWT and RELAPS, in identifying ATTR cardiac amyloidosis, detected by DPD scintigraphy. On the other hand, MW was modestly related to biomarkers such as Troponin and NT-pro-BNP over and above RWT and RELAPS.

Results interpretation: This study illustrates additional application of MW in the assessment of patients with heart failure. As MW incorporates both LV deformation and afterload, measured as blood pressures, it might reflect ventricular arterial coupling and therefore is potentially of interest in identifying different phenotypes and prognosis in HF. MW has previously been proposed mainly as a marker of LV dys-synchrony, although on a limited scale. [16] Few studies have investigated the use of MW in cardiac amyloidosis. However, recent studies showed that MW can predict mortality in CA in a similar level as GLS [8,17]. Another study found MW in CA to be related to myocardial external efficiency and further reduced with exercise [18]. However, these studies included AL-CA, which we did not.

Our study assessed the impact of myocardial ATTR on LV cavity MW compared with control patients with other underlying pathologies causing increased wall thickness and have demonstrated that the former pathology greatly suppresses MW compared to the latter. Interestingly, these findings were not independent of the overall cardiac performance. Indeed, MW was related to the pro-BNP levels and to a greater extent to the troponin levels, as has been shown by Roger-Rolle et al. [8]. The mechanisms behind such relationships are not similar, since pro-BNP reflects the rise in myocardial wall stress and troponin reflects myocyte damage. Despite the difference in mechanisms, the relationship we found, irrespective of its strength, suggests that the two mechanisms potentially exist in ATTR-CA. In addition to this finding was the higher mortality we found in the ATTR-CA patients compared to the non-amyloid patients, as has been shown by Clemmensen et al. [17].

In contrast to the relationship between MW and cardiac endocrine and myocyte function, our results showed no relationship between MW and other LV structural function parameters in the form of RWT and RELAPS. This finding was not surprising based on the above discussed mechanisms. Importantly, the difference in MW between the two groups disappeared after excluding those with atrial fibrillation, which is a common arrhythmia in ATTRwt and is also known to reduce long axis function (GLS). This strengthens the finding that MW seems not useful in differentiating infiltrative from other causes of thickened LV wall. Finally, the failure of LV MW in predicting the diagnosis of ATTR-CA and clinical prognosis was not a complete surprise since MW reflects massive global dysfunction while diagnosis of ATTR-CA by DPD could depend on only mild segmental infiltration with minimal global function implication. As for mortality, it is a complex clinical end-product with too many factors implicating it, so unlikely for a single marker of ventricular performance to accurately predict it.

Limitation: This study has limitations. The studied number of patients and controls was modest, justified by the rarity of the ATTR-CA among heart failure centers. Patients with HFnCA can be considered heterogeneous, so analyzing them as small subgroups was not scientifically feasible. We were unable to assess the rate of ATTR-CA progression, since patients did not have regular DPD follow up scans with, even, subjective assessment of the extent of myocardial infiltration. Calculation of MW involves multiple parameters including fluctuating blood pressure measurements; thus a small error is likely to be significantly exaggerated and impact the overall statistics and conclusion. Myocardial work calculation was not feasible in all patients which was mainly explained by lack of accurate image quality from all three projections. Finally, no cardiac biopsy was taken to confirm the final diagnosis.

Clinical implications: Although proposed as a potentially useful marked of LV cavity performance in different conditions, LV MW has proved to be of little additional value in identifying ATTR-CA disease and its prognosis, compared with conventionally, well tested GLS measurements.

Conclusion: Myocardial work failed to provide additional clinical value over and above RWT or RELAPS in identifying ATTR-CA and in predicting its prognosis. It is, however, related to biomarkers reflecting raised LV filling pressures and myocardial damage.

## Figures and Tables

**Figure 1 jcm-10-04555-f001:**
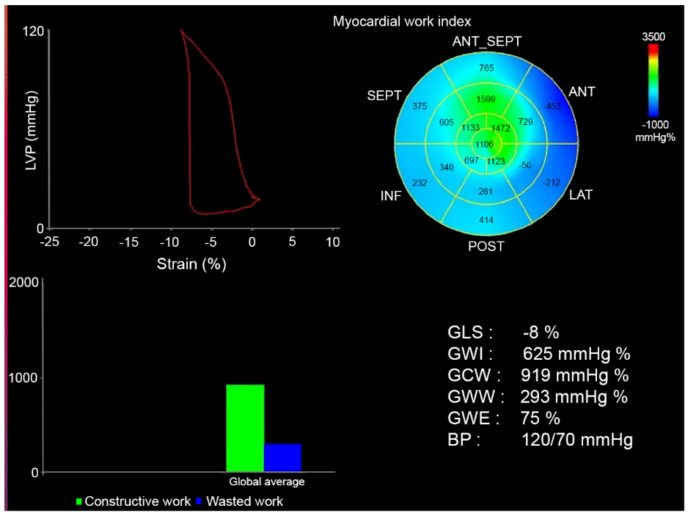
Myocardial work analysis in a patient with transthyretin cardiac amyloidosis (ATTR-CA). Green color in the Bull’s eye and graph lower left represents well preserved myocardial work (apical section) and blue color wasted work (base and mid ventricular segment, this in accordance with apical sparing pattern. The Y-axis in the lower left represents global constructive (GCW) and wasted work (GWW) (mmHg/%).

**Figure 2 jcm-10-04555-f002:**
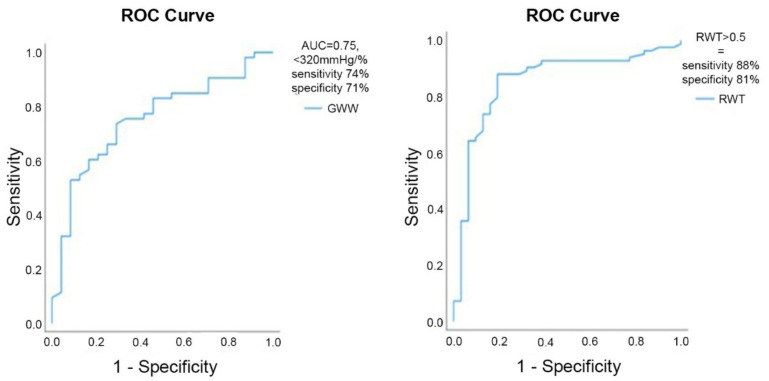
Left curve: ROC analysis from GWW (AUC = 0.75) identifying ATTR-CA and right curve: RWT (AUC = 0.84) identifying ATTR-CA.

**Table 1 jcm-10-04555-t001:** Clinical data at the time for DPD verified diagnosis.

	ATTR-CA (*n* = 86)	HFnCA (*n* = 30)	*p*-Value
NT-proBNP log, ng/L	3.1 ± 0.6	2.9 ± 0.7	0.007
Troponin, ng/L	35 (36)	23 (24)	0.148
SBP, mmHg	130 ± 20	142 ± 18	0.020
DBP, mmHg	80 (14)	85 (17)	0.004
Height, cm	175 ± 8	173 ± 9	0.201
Weight, kg	78 ± 13	87 ± 21	0.015
BMI, kg/m^2^	25 ± 4	29 ± 6	<0.001
Female gender (%)	19 (22%)	11 (35%)	0.06

ATTR-CA: transthyretin cardiac amyloidosis, HFnCA: heart failure with no cardiac amyloidosis, SBP: systolic blood pressure, DBP: diastolic blood pressure.

**Table 2 jcm-10-04555-t002:** Echocardiographic data in ATTR-CA and HFnCA in all patients.

	ATTR-CA (86)	HFnCA (30)	*p*-Value ATTR-CA vs. HFnCA
Age, years	77 (42)	75 (8)	0.712
HR, bpm	72 ± 12	66 ± 13	0.05
IVS, mm	18 (5)	15 (2)	0.005
LVDD, mm	44 ± 5	48 ± 7	0.004
PWT, mm	14.0 ± 2.9	10.5 ± 1.8	<0.001
LAVI, mL/m^2^	39 (18)	38 (17)	0.943
RWT, mm	0.65 ± 0.18	0.45 ± 0.13	<0.001
RV-RA, mmHg	25 (10)	25 (10)	0.321
E velocity, cm/s	80 (22)	60 (30)	0.002
RELAPS	2.2 (1.0)	0.7 (0.4)	<0.001
AF, %	27	3	<0.001
GLS, %	12.7 ± 3.9	12.6 ± 4.5	0.910
GWI, mmHg%	1180 ± 473	1453 ± 647	0.040
GCW, mmHg%	1560 ± 547	1908 ± 772	0.026
GWW, mmHg/%	210 (89)	380 (268)	<0.001
GWE, %	82 (9)	83 (13)	0.0497

HR-heart rate, IVS-interventricular septum, LVDD-left ventricular diastolic diameter, PWT-posterior wall thickness in diastole, LAVI-left atrial indexed volume, RWT-relative wall thickness, GLS-Global Longitudinal Strain, RV-right ventricular, RA-right atrial, E-early diastolic, RELAPS-relative apical sparing, GCW-Global Constructive Work GWW-Global Wasted Work, GWE-Global Work Efficiency; AF-atrial fibrillation, *p* < 0.05 comparing groups.

**Table 3 jcm-10-04555-t003:** Echocardiographic data in ATTR-CA and HFnCA in patients with sinus rythm.

	ATTR-CA (28)	HFnCA (22)	*p*-Value ATTR-CA vs. HFnCA
GLS, %	14.1 ± 2.9	12.4 ± 4.5	0.102
GWI mmHg%	1446 ± 384	1449 ± 668	0.984
GCW mmHg%	1865 ± 440	1876 ± 791	0.985
GWW mmHg%	319 ± 190	409 ± 173	0.091
GWE, %	83 ± 7	80 ± 7	0.118
RELAPS	1.9 (1.1)	0.8 (0.73)	<0.001
RWT, mm	0.65 ± 0.17	0.46 ± 0.13	<0.001

GLS-Global lLngitudinal Strain, GWI-Global Work Index, GCW-Global Constructive Work GWW-Global Wasted Work, GWE-Global Work Efficiency, RELAPS-relative apical sparing, RWT-relative wall thickness.

**Table 4 jcm-10-04555-t004:** Relation between myocardial work and biomarkers in all patients and in ATTR-CA.

	NT-proBNP All Patients	Troponin All Patients	NT-proBNP ATTR-CA	Troponin ATTR-CA
GLS	R = −0.18, *p* = ns	R = −0.38, *p* = 0.001	R = −0.28, *p* = 0.044	R = −0.47, *p* < 0.001
GWI	R = −0.27, *p* = 0.019	R = 0.40, *p* = 0.001	R = −0.37, *p* = 0.006	R = −0.44, *p =* 0.001
GCW	R = −0.24, *p* = 0.038	R = −0.35, *p* = 0.002	R= −0.38, *p* = 0.005	R = −0.41, *p* = 0.003
GWW	R = −0.12, *p* = ns	R = −0.21, *p* = 0.067	R = −0.085, *p* = ns	R = −0.140, *p* = ns
GWE	R = −0.34, *p* = 0.003	R = −0.22, *p* = 0.054	R = −0.085, *p* = ns	R = −0.134, *p* = ns
RELAPS	R = 0.19, *p* = ns	R = 0.22, *p* = 0.026	R = 0.21, *p* = ns	R = 0.17, *p* = ns
RWT	R = 0.26, *p* = 0.006	R = 0.26, *p* = 0.006	0.27, *p* = 0.014	R = 0.21, *p* = 0.070

GLS-Global Longitudinal Strain, GWI-Global MW Index, GCW-Global Constructive Work GWW-Global Wasted Work, GWE-Global Work Efficiency, RELAPS-relative apical sparing, RWT-relative wall thickness.

**Table 5 jcm-10-04555-t005:** Myocardial work and death in ATTR-CA.

		N	Mean/Median *	SD/IQR *	*p*-Value
GLS	Dead	19	11.3 *	4.0 *	0.043 *
	Alive	34	13.5 *	3.6 *	
GWI	Dead	19	977 *	462 *	0.018 *
	Alive	34	1295 *	777 *	
GCW	Dead	19	1301	543	0.009
	Alive	34	1705	501	
GWW	Dead	19	209 *	169 *	0.911 *
	Alive	34	221 *	211 *	
GWE	Dead	19	83 *	12 *	0.043 *
	Alive	34	83 *	11 *	
RWT	Dead	30	0.66	0.26	0.039
	Alive	54	0.57	0.30	
RELAPS	Dead	28	2.4 *	1.4 *	0.856 *
	Alive	52	2.0 *	1.2 *	

GLS-Global Longitudinal Strain, GWI-Global MW Index, GCW-Global Constructive Work GWW-Global Wasted Work, GWE-Global Work Efficiency, RELAPS-relative apical sparing, RWT-relative wall thickness. *: median/IQR.

## Data Availability

The data presented in this study are available on request from the corresponding author. The data are not publicly available due to ongoing research analysis.

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
