# Peer review of "Myocardial Work Does Not Have Additional Diagnostic Value in the Assessment of ATTR Cardiac Amyloidosis"

_jcm, 2021, doi:10.3390/jcm10194555_

Round 1

Reviewer 1 Report

The paper ”Myocardial work does not have additional diagnostic value in the assessment of ATTR cardiac amyloidosis” aims to investigate if myocardial work assessed from strain-pressure loops obtained by echocardiography can differentiate thransthyretin cardiac amyloidosis (ATTR-CA) from heart failure with increased septal thickness with no cardiac amyloidosis. A secondary aim was to test the accuracy of myocardial work in predicting the clinical prognosis in ATTR-CA. The authors conclude that myocardial work is unlikely to have additional value in improving the diagnosis of heart failure in ATTR-CA.

Major comments:

1. The results of the study, and thus the conclusions, are highly dependent on the strain analysis. As described by Smiseth et al. in “How to measure left ventricular myocardial work by pressure–strain loops” (EHJ- Cardiovasc Imag 2021), the positioning of the region of interest will affect the resulting myocardial work. Intra-and interobserver analysis of key variables (i.e. strain) used for calculation of myocardial work should be provided. Also, please cite this reference in methods.

2. It is not described in the method section how blood pressure was determined.

3. When was blood pressure measured?  Was it parallel to the image acquisition used to calculate strain? If not, please discuss how this may have affected measurements (for example if heart rate differed between blood pressure measurement and echo.

4. Figure 1. This figure is somewhat confusing as to how it contributes to the paper, it might be superfluous. If kept, please indicate in the legends if/how the 17 segment model shown is used in the paper. Furthermore the y axis graph at the bottom left lacks label and units. Also, please increase the size of the text in the figure. It is currently a lot of empty black space and very small letters.

5. Discussion, line 205 Results interpretation: I do not understand how this interpretation and reference relates to the study findings. Please clarify.

6. Differences between groups disappeared when only subjects in sinus rhythm were assessed. What is the mechanism behind this? Can results from pressure-strain curves be trusted when patients are in Afib? Please elaborate.

7. There were more women in the ATTR group compared to HFnCA. Sex affects blood pressure, and blood pressure in turn affect myocardial work. Are the results we see in the study affected by sex differences between groups?

8. Myocardial work was only feasible in 24 of 30 HFnCA and 53 of 86 ATTR. Please explain why

9. Line 208-214 in discussion. Please clarify which reference you are referring to. For example: “Few studies have investigated the use of MW in cardiac amyloidosis but one showed that MW predicted mortality similar to GLS (8, 16).” Which one is the “the one” ? The results of reference 8 seem to be opposite of the present study by claiming MW can predict mortality. If I may speculate, I suspect reference 8 triggered the writing of the current paper– please elaborate on why the results differ between studies. Isn’t this the most important message of the study?

Minor comments;

  1. Line 126, please explain abbreviation FDPD. However, I believe this is the only time it’s used in the whole paper and therefore suggest the authors avoid this abbreviation all together.
  2. Table 2, 3 and 4. Please check tables carefully for missing spaces, additional spaces and spelling/ use of capital letters or not in legends. For example, Table 2 RV= Right ventriculat should be right ventricular, Table 3 GCW= Global Constructive work should be Global Constructive Work.
  3. Line 172. Referring to figure 1, should be figure 2.
  4. Avoid starting a new sentence with an abbreviation, for example line 181 ; MW should be Myocardial work. Also for line 188.
  5. Figure 2, right panel. The lable is RWT2 – why is there the figure 2?
  6. Table 5. I suggest to use the word “deceased” rather than ”dead”.
  7. Table 5. The column SD/IQR is confusing – is it SD or is it IQR. Please clarify.

Author Response

Dear editor at JCM

We are grateful to have the possibility to respond to the major issues raised by the reviewers. We believe that the manuscript now has improved after the adjustments recommended. All changes in text are highlighted in red.

Reviewer 1:

Major comments:

  1. The results of the study, and thus the conclusions, are highly dependent on the strain analysis. As described by Smiseth et al. in “How to measure left ventricular myocardial work by pressure–strain loops” (EHJ- Cardiovasc Imag 2021), the positioning of the region of interest will affect the resulting myocardial work. Intra-and interobserver analysis of key variables (i.e. strain) used for calculation of myocardial work should be provided. Also, please cite this reference in methods.

Response: We have followed the recommendation from Smiseth et al and reference 14. Interobserver variability (ICC) is added at the end of the results. The ICC between two readings were 0.96.

  1. It is not described in the method section how blood pressure was determined.

Response: Blood pressure was taken using conventional cuff Blood pressure mercury manometer, after the echocardiography examination and a mean of three consecutive measurements was taken.

  1. When was blood pressure measured?  Was it parallel to the image acquisition used to calculate strain? If not, please discuss how this may have affected measurements (for example if heart rate differed between blood pressure measurement and echo.

Response: See Q2

  1. Figure 1. This figure is somewhat confusing as to how it contributes to the paper, it might be superfluous. If kept, please indicate in the legends if/how the 17 segment model shown is used in the paper. Furthermore the y axis graph at the bottom left lacks label and units. Also, please increase the size of the text in the figure. It is currently a lot of empty black space and very small letters.

Response: The figure has been now improved and we agree it might be considered superfluous. However, we have added more text to the figure and therefore kept it.

  1. Discussion, line 205 Results interpretation: I do not understand how this interpretation and reference relates to the study findings. Please clarify.

Response: We agree that this statement was not accurate, specially at the beginning of interpretation. This has now been moved but kept as LV dys-synchrony assessment is an important issue in the use of myocardial work technique.

  1. Differences between groups disappeared when only subjects in sinus rhythm were assessed. What is the mechanism behind this? Can results from pressure-strain curves be trusted when patients are in Afib? Please elaborate.

Response: It is known that atrial fibrillation by itself reduces long axis function, particularly when it is fast. Therefore it is relevant to explain the difference in GLS by the increased prevalence of AF in ATTR-CA.

  1. There were more women in the ATTR group compared to HFnCA. Sex affects blood pressure, and blood pressure in turn affect myocardial work. Are the results we see in the study affected by sex differences between groups?

Response: We are grateful for this observation. There were fewer females in both groups. Please see table with more data.

  1. Myocardial work was only feasible in 24 of 30 HFnCA and 53 of 86 ATTR. Please explain why.

Response: This is explained by limitation in aquiring accurate image quality from 3 planes. (ie. lack of lateral resolution and foreshortening)

  1. Line 208-214 in discussion. Please clarify which reference you are referring to. For example: “Few studies have investigated the use of MW in cardiac amyloidosis but one showed that MW predicted mortality similar to GLS (8, 16).” Which one is the “the one” ? The results of reference 8 seem to be opposite of the present study by claiming MW can predict mortality. If I may speculate, I suspect reference 8 triggered the writing of the current paper– please elaborate on why the results differ between studies. Isn’t this the most important message of the study?

Response: Both the study by Clemmensen and Roger-Rolle found myocardial work being prognostic for all-cause mortality but in a similar level as GLS. This is now changed. Our results are consistent with these observations. However, we did not calculate AUC for predicting all cause mortality.

Minor comments;

  1. Line 126, please explain abbreviation FDPD. However, I believe this is the only time it’s used in the whole paper and therefore suggest the authors avoid this abbreviation all together.

Response: This is an erratum, it should be DPD. It has now been corrected.

  1. Table 2, 3 and 4. Please check tables carefully for missing spaces, additional spaces and spelling/ use of capital letters or not in legends. For example, Table 2 RV= Right ventriculat should be right ventricular, Table 3 GCW= Global Constructive work should be Global Constructive Work.

Response: Thank you for this observation, we have now changed this.

  1. Line 172. Referring to figure 1, should be figure 2.

Response: This has been corrected.

  1. Avoid starting a new sentence with an abbreviation, for example line 181 ; MW should be Myocardial work. Also for line 188.

Response: This has been corrected

  1. Figure 2, right panel. The lable is RWT2 – why is there the figure 2?

Response: This has been changed

  1. Table 5. I suggest to use the word “deceased” rather than ”dead”.

Response. Thank you. This has been changed

  1. Table 5. The column SD/IQR is confusing – is it SD or is it IQR. Please clarify.

Response: Thanks for the comment. This is also now changed.

Reviewer 2:

Lines 15, 55, 61, 65, and 155: Please clarify septal thickness cut-off >14mm or >=14mm.

Response: This is now changed to >= 14mm.

61: Were biopsy results available for any of the amyloidosis patients that were positive by DPD and ruled out for AL-CA? Please specify.

Response: No cardiac biopsy was done to confirm diagnosis of AL-CA. This is now stated in the limitations section.

157-159: Was cardiac MRI done in patients labelled as HCM? Please specify.

Response: No MRI was done to confirm HCM diagnosis.

160: Specify which value of "blood pressure" - systolic or diastolic or mean or combination.

Response: This is now changed in the results. Systolic and diastolic….

161: In addition to "weight", BMI would be a more valuable metric. Please specify.

Response: BMI is now added in table 1.

Table 1: Add p-value for BMI in addition to Ht and Wt.

Response. See above.

Table 2: Provide Left Atrial Volume Index instead of LAV. Also was Strain rate available in addition to GLS? If so, provide data.

Response: This is now changed in table 2.

Table 5: Provide specifics for mortality - clarify "all cause mortality", and from what sources was this data obtained.

Response: Now changed to all cause mortality.

Reviewer 2 Report

In this relevant and interesting study, the authors sought to determine the

Lines 15, 55, 61, 65, and 155: Please clarify septal thickness cut-off >14mm or >=14mm.

61: Were biopsy results available for any of the amyloidosis patients that were positive by DPD and ruled out for AL-CA? Please specify.

157-159: Was cardiac MRI done in patients labelled as HCM? Please specify.

160: Specify which value of "blood pressure" - systolic or diastolic or mean or combination.

161: In addition to "weight", BMI would be a more valuable metric. Please specify.

Table 1: Add p-value for BMI in addition to Ht and Wt.

Table 2: Provide Left Atrial Volume Index instead of LAV. Also was Strain rate available in addition to GLS? If so, provide data.

Table 5: Provide specifics for mortality - clarify "all cause mortality", and from what sources was this data obtained.

Author Response

(The authors gave the same response as above.)
